# Secure and Anonymous Voting D-App with IoT Embedded Device Using Blockchain Technology

Cristian Toma *, Marius Popa, Catalin Boja, Cristian Ciurea and Mihai Doinea

Department of Economic Informatics and Cybernetics, Bucharest University of Economic Studies, 010552 Bucharest, Romania; marius.popa@ie.ase.ro (M.P.); catalin.boja@ie.ase.ro (C.B.); cristian.ciurea@ie.ase.ro (C.C.); mihai.doinea@ie.ase.ro (M.D.)
* Correspondence: cristian.toma@ie.ase.ro

**Abstract:** The paper presents the construction of a proof-of-concept for a distributed and decentralized e-voting application in an IoT embedded device with the help of blockchain technology. A SoC board was used as an IoT embedded device for testing the PoC. This solution ensures complete voter anonymity and end-to-end security for all entities participating in the electronic voting process. The paper outlines the solution's two layers. Implementation details are presented for the e-voting application, which was deployed inside of an IoT embedded device. The solution and presented protocols provide two major properties: privacy and verifiability. To ensure privacy, the proposed solution protects the secrecy of each electronic vote. As for implementing verifiability, the solution prevents a corrupt authority from faking in any way the process of counting the votes. Both properties are achieved in the presented solution e-VoteD-App.

**Keywords:** blockchain; e-VoteD-App; decentralized application; IoT—Internet of Things; cyber security; mobile-embedded; secure element





## 1. Introduction

As considered in [1], the voting process is a very non-transparent mechanism in many cases, and it is known to be exposed to different acts of corruption. The introduction of blockchain as a service in the voting process allows for creating a protocol so that the voting process is completely open, fair, and independently verifiable by everyone. Different electronic voting mechanisms have been applied in countries and regions such as Estonia, Australia, India, and Malaysia.

Because voting systems are subject to attacks while discussing democratic decision-making, a solution for a trustless electronic voting platform is provided by leveraging blockchain technology [2].

The blockchain protocol is a way to log and verify records that is transparent and distributed among users. An e-voting system based on blockchain could accelerate, simplify, and reduce elections costs and might even lead to higher voter turnouts and the development of stronger democracies [3].

Another advantage provided by e-voting systems consists in increasing the efficiency and reducing errors during the voting process. To improve the overall resilience of such systems, the blockchain mechanism is needed to achieve an end-to-end verifiable e-voting scheme [4].

In addition to ensuring security, an e-voting system should not allow duplicated votes and should be completely transparent, while protecting the privacy of the voters [5]. The introduction of blockchain technology in electronic voting systems has the advantage of improving reliability, transparency, and efficiency of the voting process [6].

The main features of blockchain technology, which make it so important to use in e-voting systems, are the immutability property and its decentralized architecture. From a practical perspective, blockchain can be successfully used to support complex applications [7].

### 1.1. Main Problem

The main problem defined by the authors was to build an e-voting system that implements the following characteristics at vote level:

- Anonymity/Privacy—the vote is not associated with any person;
- Accuracy—the vote is not able to be modified;
- Transparency/Individual and universal verifiability—any entity can verify that their vote was received and counted properly;
- Eligibility—only eligible voters can cast a vote;
- Fairness—vote results should be available only when voting session is closed.

Other characteristics such as integrity, audit, or mobility are specific properties for the underlaying system.

### 1.2. Our Contribution

This paper presents the secure e-voting schemes and implementation details of a proof-of-concept solution for a distributed and decentralized e-voting application in an IoT embedded device using blockchain technology.

This solution provides full anonymity of the voters and ensures end-to-end security for the entities involved in the electronic vote process.

The first sections are dedicated to the existing electronic voting systems and D-App voting solutions and types of blockchains used for such applications.

A general framework of voting systems with characteristics and processes that underline security issues is presented in Section 3 of the paper.

The two layers of the e-VoteD-App solution are presented in detail in the fourth section. In the first layer, the e-VUIDs (electronic Vote Unique Identifiers) are provided by using blind digital signatures to ensure the anonymity of the vote. The second layer of the solution is described in the third section of the paper, where the Ethereum (ETH) blockchain technical specifications are used in the proof-of-concept solution.

In the fifth section, the implementation details are pinpointed for the voting D-App deployed into an IoT embedded device with the secure element for crypto processing and blockchain wallet keys generation. In the last section, the authors present the conclusions, the cyber security challenges, and possibilities for future work on improving the solution using secure elements and post-quantum cryptography algorithms.

Figures 1 and 2 are a helpful summary of the voting process. Figure 2 is presenting the contribution of the paper for an electronic voting solution. Figure 2 improves upon the simplified existing voting processes from Figure 1. In this way, the proposed solution ensures complete voter anonymity and end-to-end security for all entities participating in the electronic voting process.

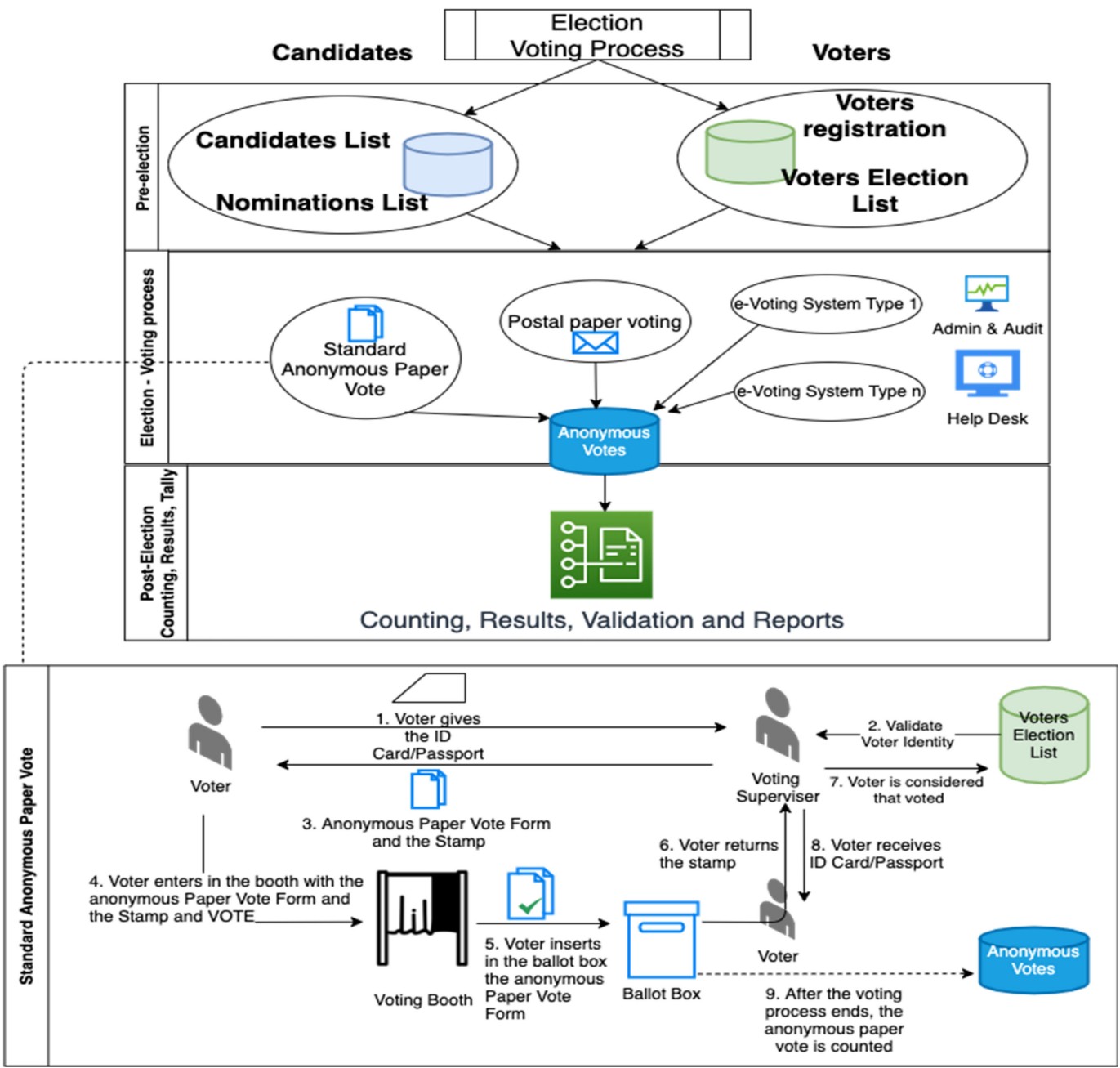

**Figure 1.** Simplified overview of the voting processes.

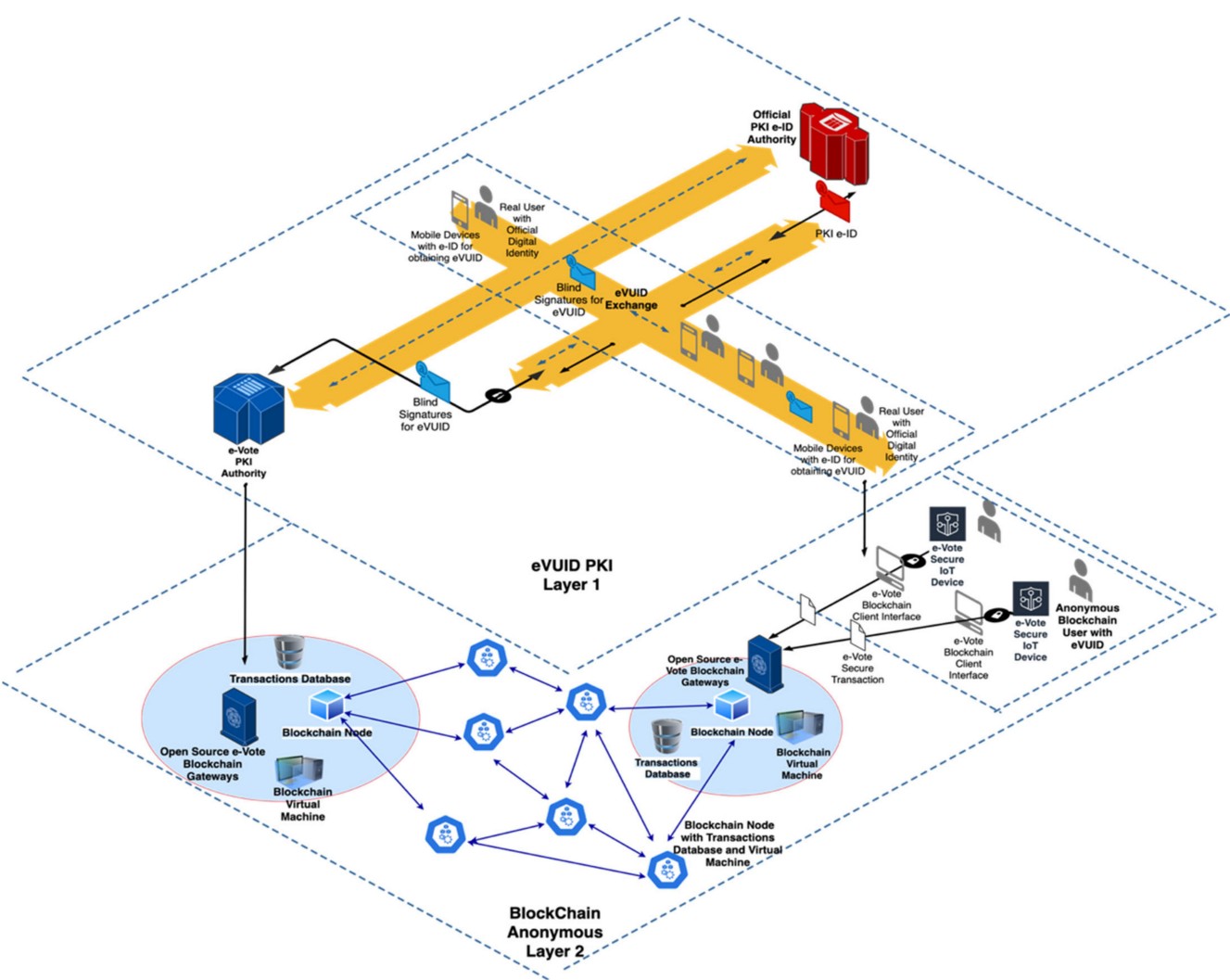

**Figure 2.** Architecture of the e-VoteD-App Solution System—2 layers.

## 2. Related Work

In [8], it is considered that electronic voting systems can be used to reduce costs. System integrity and security are guaranteed by the developments in blockchain technology, which allows for introducing a decentralized electronic voting system based on a blockchain and smart contract.

An implementation of an e-voting system that does not rely on any trusted authority to protect the voter's privacy is presented in [9]. It is based on a smart contract for Ethereum.

Another proposal for electronic voting based on blockchain is presented in [10]. It uses a digital-currency analogy, where the eligible voters can cast a ballot anonymously using a computing environment to make the public electoral process cheaper, faster, and easier. In [11], it is considered that building a secure distributed electronic voting system that offers the fairness and privacy of existing voting mechanisms, while providing the transparency and flexibility delivered by electronic systems, has been a challenge for a long period of time.

In [12], a protocol is proposed that uses blockchain technology to turn election protocol into an automated control system without relying on any single point or entity. The implementation of a proper architecture can lead to features such as data confidentiality, data integrity, and data authenticity. The pervasiveness of current e-voting systems makes them vulnerable for attacks. As Joseph Stalin is famously quoted as saying: "Those who vote decide nothing. Those who count the vote decide everything." e-Voting systems need



to assure privacy and integrity so that the balance can be shifted towards the ones who vote. Two important principles need to find their place in such systems: privacy, meaning the votes should be easily verified [13,14], without compromising voter's identity [15], and integrity, meaning once the votes are cast, the systems should not allow for them to be changed.

The university environment is a place where e-voting systems could easily be designed, tested, and implemented, since all students are keen on using instruments such as mobile and IoT embedded devices that can increase reliability and usefulness of the casting process. Attempts have been made to use electronic voting in students' elections in several universities and undergraduates' schools, as presented in [16–18].

Since e-voting systems can be used for hierarchization of a set of entities, this could mean that they can also be used in different sets of scenarios starting from informal decision-making processes up to management or government applicability [19].

## 3. Methods and Techniques in Voting Systems

The proposed solution is disruptive, and it is complementary to (not necessarily intended to replace) the existing voting solutions. The solution is inspired by the standard voting process and may be used as alternative, replacement, or complement to the existing systems. It may be deployed in companies' boards elections or other processes that involve election and voting. Figure 1 provides the overview of the voting process, which consists of several phases: Pre-election, Election, and Post-election.

In pre-election phase, the Election Authority must create the nomination and candidates list according to the laws and regulations. For the voters, the Election Authority must allow the voters registration for different types of votes (e.g., postal paper voting or e-voting System) and then create the voters election list.

The second phase of the voting system is the election per se—the voting process. The voting process may take place via (a) standard anonymous paper vote, (b) postal paper, or (c) electronic vote.

The third phase consists of counting the results from the second phase, validation processes, reports, and statistics.

The electronic vote (see Figure 1 e-Voting System Type 1 . . . n) from the second phase may involve different implementations and approaches, such as the following:

- Paper-based electronic voting systems are inspired by the standard anonymous paper vote, which is used as a system where votes are cast and counted by hand, using paper ballots. With the progress of automatic data acquisition (see QR codes) and electronic tabulation, these systems handle paper cards or sheets that are filled by hand but counted electronically.
- A direct-recording electronic (DRE) voting machine is an alternative that records votes as ballot results activated by the voters (buttons or a touchscreen); processes data with software applications; and records the voting data structures and ballot files in the memory components.
- A public network DRE (direct-recording electronic)/online/Internet voting system is an election system that uses electronic ballots and transmits the vote data from the polling place/office to another location (e.g., Central Election Authority) over a public network. These voting systems may also entail Internet, telephone/mobile equipment (ME), or embedded/IoT device voting. In the case of usage of polling locations/places/offices, the vote information may be sent as individual ballots as they are cast, periodically as batches of ballots throughout the election day, or as one batch at the close of voting. In the case of Internet voting, the voter may use his/her own location or may use the standard voting booths at polling locations using Internet-connected voting systems. For instance, at local and parliament elections in Estonia, most Estonian citizens carry a national identity card (e.g., Java Card smart card), and they vote with these cards. All a voter needs is a computer/tablet, an electronic card reader, and their ID card and its PIN, and they can vote from anywhere

in the world. Estonian electronic votes are casted during the days of advance voting. On the election day itself, people go to the polling locations/offices/stations and fill in paper ballots.

The e-voting system is inspired by the standard anonymous paper vote, and the simplified steps for the paper vote process are presented at the bottom of Figure 1, as follows:

1. The voter enters in the precinct of the election office, and he/she presents the ID card or passport to the voting supervisor in the assigned room.
2. The voting supervisor checks the voter identity and connects to the central database with the list of registered voters for voter validation and to avoid double/multiple voting.
3. The voter supervisor gives the stamp and anonymous paper vote form to the voter.
4. The voter enters in the booth with the anonymous paper vote form and the stamp to vote.
5. The voter exit from the booth and fold the anonymous paper vote form and inserts it into the ballot box.
6. The voter returns the used stamp to the voting supervisor.
7. The voting supervisor updates the list of registered voters database with the new status of the voter (e.g., vote performed); from this moment, the same voter cannot vote anymore in the same election process.
8. The voter receives back the ID card or passport.
9. At the end of the day, the ballot box is opened by the voting supervisors and the vote forms are counted according to the law and regulations. The data are sent to the Central Election Authority along with the official minutes and reports.

As an analogue to the standard anonymous paper, the e-voting system (with the name e-VoteD-App) has two layers, and it is built with several modifications, as follows:

1. The voter uses an IoT/embedded or mobile application in the first layer of e-VoteD-App issued/published by the Central Election Authority. Without any anonymity and presenting the ID card or passport in the app, the user triggers the generation of the "stamp", a unique number, as a hash blind-signed by Central Election Authority e-Vote Unique Identifiers (e-VUIDs).
2. The Central Election Authority back-end checks the voter identity and connects to the central database (e.g., via secure REST API and microservices) with the list of registered voters for voter validation and to avoid double/multiple voting.
3. The Central Election Authority back-end/cloud apps give the digitally blind-signed "stamp" to the voter application.
4. The voter enters in the second layer (blockchain layer) of the e-VoteD-App with the anonymous "Stamp" to vote. The "stamp" may be exchanged between the voters to increase anonymity of the system.
5. The voter performs the vote in the second layer of the e-VoteD-App as a blockchain transaction according to a smart contract (partially equivalent to a paper voting form) within a fully anonymous blockchain platform (e.g., Ethereum).
6. Automatically and anonymously, the "used stamp" is validated and marked as used within layer 1 at the Central Election Authority back-end/cloud apps.
7. The Central Election Authority back-end/cloud apps update the list of registered voters database with the new status of the "stamp" (e.g., "stamp"/e-VUID used); from this moment, any voter cannot vote anymore on the same election process with the same stamp.
8. The layer 2 blockchain decentralized application disables the possibility to reuse the "stamp"/e-VUID in the blockchain transaction according to the smart contract.
9. At the end of the day, the "ballot box" (blockchain hyperledger) is opened by the Central Election Authority layer 2 blockchain decentralized application, to have access to all blockchain transactions for the electronic vote according to the smart contract. The vote transactions are validated against anonymous e-VUIDs and they are counted

according to the modified law and regulations. The data are sent to the Central Election Authority along with the official minutes and reports.

The blind signature together with e-VUIDs and the first layer of the proposed architecture provide the anonymity of the solution.

The proposed system may be scaled up for the national and international level or scaled down for companies or different business elections requirements.

## 4. Architecture and Cryptographic Schemes of e-VoteD-App

The solution PoC (proof-of-concept) uses IoT devices, blind signatures [20], Ethereum blockchain, and smart contracts [21].

There are two layers in the architecture of the e-vote solution:

- Layer 1 (Top) is responsible for voters' generation of the e-VUIDs (e-Vote Unique Identifiers) necessary for an anonymous ballot process. It is based on the blind signatures, and mainly, the voters generate 32 bytes VUID and obtain the blind signatures (RSA 2048 bits) from the e-vote authority.
- Layer 2 (Bottom) is fully based on blockchain technology, and the users from the first layer use e-VUIDs in an anonymous and secure manner.

### 4.1. First Layer of the Architecture for Anonymity in the e-VoteD-App Solution

In this section, the necessary computational cryptographic techniques are described for providing e-vote anonymity before using blockchain technology. The architecture of the e-vote solution system is in Figure 2, and the layers are presented in the 3D diagram.

Before jumping into the first layer from the top of Figure 2 (eVUID PKI Layer), some aspects of public key cryptography such as cryptographic key management, digital signatures, identification, encryption, and decryption need to be mastered.

There are many classes of asymmetric algorithms, but here are the main three that are used: integer factorization (e.g., RSA 2048 bits keys), elliptic curves (e.g., ECDSA with curve secp256k1), and discrete logarithm (e.g., Ephemeral DH).

The e-VoteD-App solution uses in the first layer the blind signature [20] invented by David Chaum. The basic principle is explained in Figure 3:

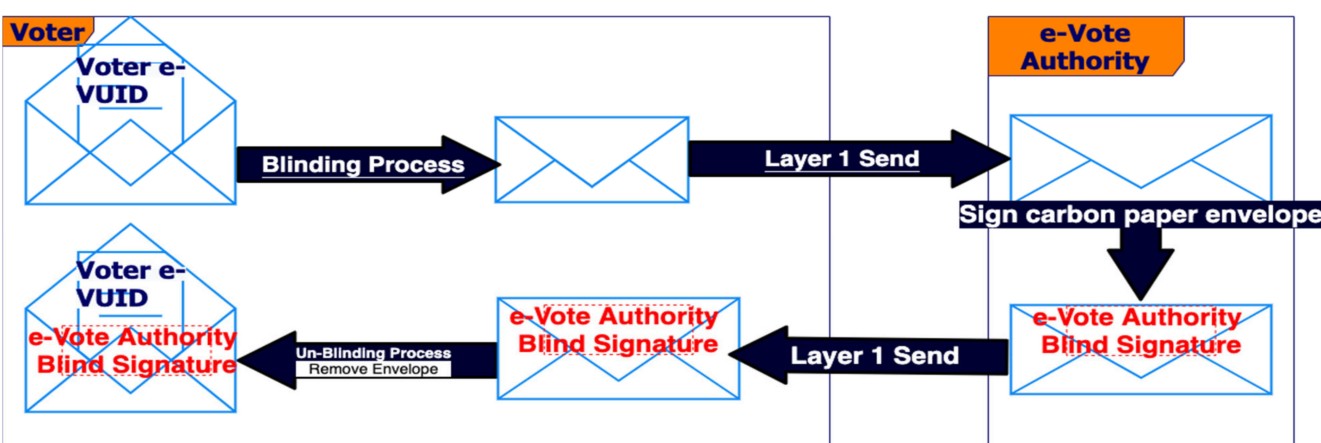

**Figure 3.** Blind Signature mechanism applied in Layer 1 of the e-VoteD-App Solution.

Blind signatures are used extensively in electronic cash systems, because they are a practical way to provide anonymity. The principle says that if someone wants to blind-sign a message, then the letter (clear message) should be put by the sender in a postal envelope made from carbon paper. The receiver signs the postal envelope made from the carbon paper, and the signature is now also on the letter (clear message). The letter is received by the receiver in the envelope. The receiver removes the carbon paper envelope and discovers the signature on the letter (clear message).

The mathematical proof for blind signatures [20] is represented in the following steps adapted for the e-voting solution:

- Sender **V (voter)** chooses a blinding factor *k* **(with gcd(k, n) = 1)**, as a random number between 1 and n. Then, the message **M** (e-vote UID or blockchain address) is blinded by computing (**e** = exponent of the public key of **A**—Authority for the e-vote and n = public modulus of **A**):

$$T = M * k^e \ (mod \ n) \tag{1}$$

- The receiver **A** signs *T* (**d** = exponent of the secret key of receiver **A**):

$$T^d = (M * k^e)^d \ (mod \ n) = (M)^d * k \ (mod \ n) \tag{2}$$

- The sender **V** receives **T^d** and unblinds it by computing:

$$S = (T^d * k^{-1}) = \left( (M)^d * k * k^{-1} \right)(mod \ n) = M^d (mod \ n) \tag{3}$$

Considering the voting solution and that the user V generates a voting e-VUID—Unique Identifier—the generic message called M here is a simple numeric example for the RSA with blinding mechanism [1,20]. It is considered the value of **M = 2**, the RSA public key of the e-vote solution authority $\mathbf{PUB}_A = (\mathbf{11, 14}) = (\mathbf{EXP_{PUB_A}, MOD_A})$, and the **private key of A⁻PRIV**$_A$ **= (5, 14)** = (**EXP_PRIV**$_A$, **MOD**$_A$) .

- The user **V** will "blind" this message with the random k = 3 (with condition gcd $(3, 14) = 1$ and the modular multiplicative inverse of 3, which is $3^{-1}$ *mod* $14 = 5 \iff 3 * 3^{-1}$ *mod* $14 = 3 * 5$ *mod* $14 = 1$) => the "blinded" value: $2 * 3^{11}$ *mod* $14 = 2 * 177147$ *mod* $14 = 354294$ *mod* $14 = 10$; therefore, T obtained is **T = 10**.
- The e-vote solution authority (user A) receives **T = 10** and will never know the real message M = 2. The authority A will apply the RSA private key $PRIV_A = (5, 14) \Rightarrow 10^5$ *mod* $14 = 100000$ *mod* $14 = 12$, *which is the signed T*—**(T^d) = 12**.
- When the voter **V** receives the signature of the e-vote solution authority, then V will extract from the **signed T** the unblinded signature by computing: **S = 12 ∗ 3⁻¹ *mod* 14 = 12 ∗ 5 *mod* 14 = 60 *mod* 14 = 4**.
    - As validation, the voter sees if $2^5$ *mod* $14 = 4$ ? It is true, therefore, that it is as if we would have sent the value M = 2 without any blinding to the user A—the e-voting authority—in order to sign the user V message with A's RSA private key.

It is forbidden for the e-vote authority to store the link between the real user and the emitted blind signature value. However, to avoid any issues with the authority storage of the signatures, the voters can interchange their e-VUIDs according to an internal set of rules—protocol.

For layer 1, the protocol between the e-vote authority and the voter is detailed in Figure 4.

A PKI—Public Key Infrastructure—needs to be implemented at the level where the e-vote process is running, this being an assumption needed for layer 1 of Figure 4 to work.

In layer 1, the authors assumed the existence at a national level or company/institution/company level (depending on where the e-vote is implemented) of a PKI—Public Key Infrastructure. There are 6 messages exchanged in 3 submissions and 14 steps self-explained in Figure 3 in order to obtain the **e-VUID** and the signature—**Signed_e-VUID**—of the e-vote authority over it, using an RSA asymmetric key algorithm with 2048 bits key length. Once the voter has an e-VUID and signed_e-VUID, then he/she may exchange these items with other voters.

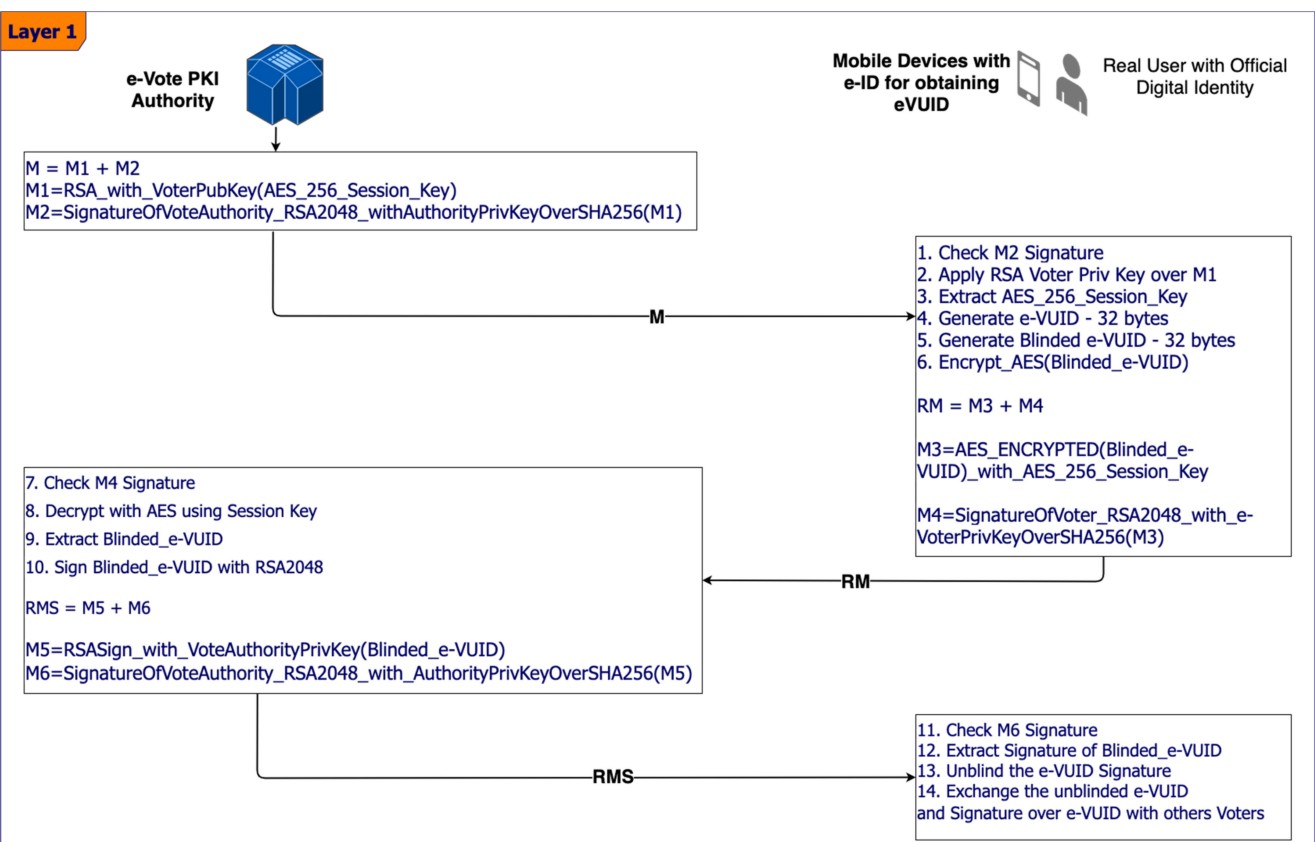

**Figure 4.** Protocol between e-Vote Authority and the voters, including the blinding mechanism.

### 4.2. Second Layer of the Architecture for the e-Voting Solution Using Blockchain

The second layer is operating with blockchain technology, and it is based on the anonymity of the **e-VUID** and the **Signed_e-VUID**.

The Blockchain is a collection of transactions and assets that allows the transfer in a secure manner of units of ownership.

Cryptography is the core component of the blockchain, and asymmetric cryptography provides security for the system. In practice, for the blockchain signatures of transactions, ECC cryptography is used [22,23].

These are well-known concepts in the public key cryptography theory and practice. In the public key cryptography signature scheme, the sender signs the data by using its private key. The signature is transmitted over the communication channel. The receiver uses the sender's public key to get clear signature to verify it against the clear data to validate the data integrity.

The reasoning behind ECC that is used for the presented proof-of-concept is detailed more in [24,25]. In the Ethereum blockchain for implementing layer 2 of the e-vote solution, while using smart contracts for voting, the voter will anonymously sign every transaction hash. The authors' PoC of the e-voting solution uses Ethereum blockchain with ECDSA signatures with the elliptic curve of standard SEC2 secp256k1. An elliptic curve is described by an equation that generates a curve over a finite field. In this phase of PoC, the solution is generating blockchain addresses using ECMAScript, as shown in Figure 5.

The authors imported the private keys into the browser Metamask wallet (www. metamask.io—accessed on 15 January 2022), but as best practice, it would be safer to interconnect with a hardware secure element (e.g., Trezor or a secure element with Java Card) with at least FIPS/CC—Common Criteria EAL 4+ security certification level.

Figure 6 shows how the address is obtained from mnemonics instead of private number, as presented in Figure 5.

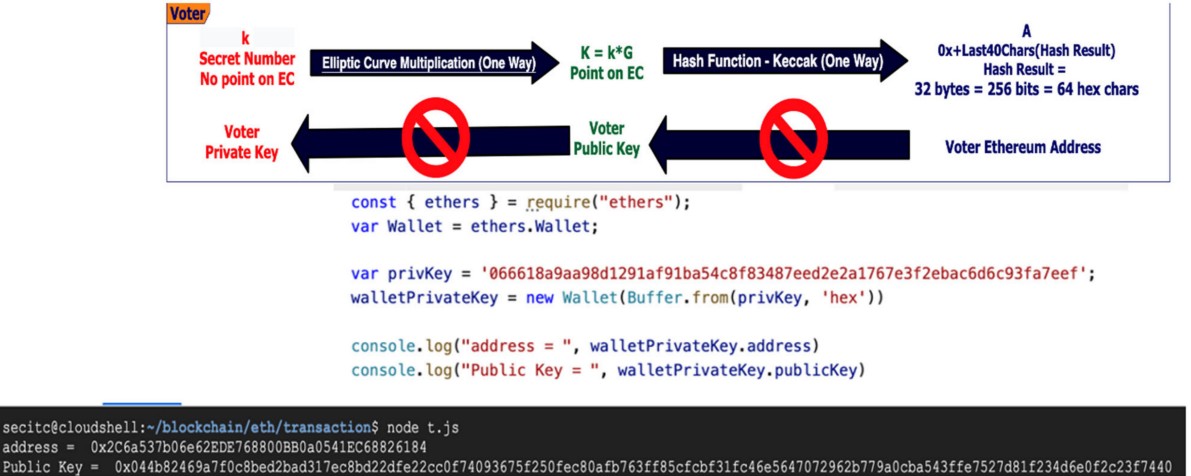

```
const { ethers } = require("ethers");
var Wallet = ethers.Wallet;

var privKey = '066618a9aa98d1291af91ba54c8f83487eed2e2a1767e3f2ebac6d6c93fa7eef';
walletPrivateKey = new Wallet(Buffer.from(privKey, 'hex'))

console.log("address = ", walletPrivateKey.address)
console.log("Public Key = ", walletPrivateKey.publicKey)
```

```
secitc@cloudshell:~/blockchain/eth/transaction$ node t.js
address =  0x2C6a537b06e62EDE768800BB0a0541EC68826184
Public Key =  0x044b82469a7f0c8bed2bad317ec8bd22dfe22cc0f74093675f250fec80afb763ff85cfcbf31fc46e5647072962b779a0cba543ffe7527d81f234d6e0f2c23f7440
```

**Figure 5.** Ethereum Address generation using node.js ethers.io module using private key.

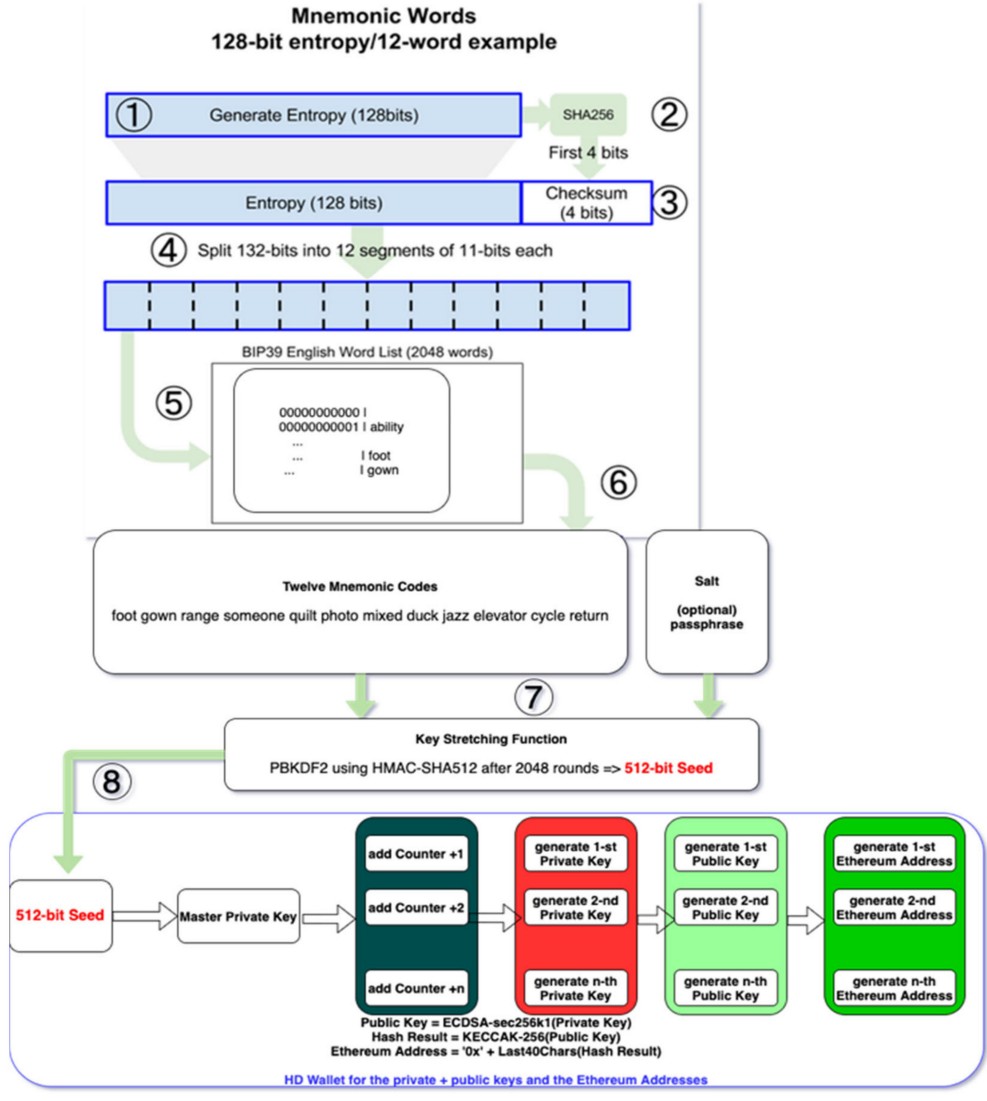

**Figure 6.** Ethereum Addresses generation from 12 words mnemonics BIP-39 "foot gown range someone guilt photo mixed duck jazz elevator cycle return" and obtaining the Ethereum Addresses within a HD Wallet.

The process of obtaining the Ethereum addresses within a HD (Hierarchical Deterministic) wallet using mnemonics involves the steps from Figure 6:

- The user/system generates the entropy from fingers movement/tapping on the screen or buttons, system time, etc., as a number of 128 bits—step 1;
- SHA-256 is applied to the entropy, and the first 4 bits from the result are concatenated with initial entropy number—step 2 and 3;
- The 132-bits number is split into 12 segments of 11 bits each—step 4;
- According to BIP39 English Word List standard, the twelve mnemonic codes are extracted, and the end-user must store them securely if the (IoT) device is destroyed—step 5;
- These twelve codes (words) together with a Salt, which optionally may be controlled by a passphrase, are the input for the Key Stretching Function—step 6;
- The Key Stretching Function is a PBKDF2 (Password-Based Key Derivation Function) using HMAC-SHA512 (Hash-based Message Authentication Code with Secure Hash Algorithm and output on 512 bits). After 2048 rounds a 512-bits seed is obtained—step 7;
- The 512-bits seed is the input for the Master Private Key. If the HD wallet stores n Ethereum addresses, then it will store n private numbers and public keys. For each Ethereum address the HD wallet generates from the master key in a deterministic way the n private keys by adding a counter—step 8;
- For each private key (number), a public key on the elliptic curve secp256k1 is generated. From each public key, an Ethereum address for the e-vote process is generated, as shown in Figure 6.

Starting from the seed (see Figures 5 and 6), an Ethereum HD wallet will contain the private and public key based on whether an Ethereum address can be generated.

It would be a security breach if someone would obtain the mnemonic from the private key. No matter how the Ethereum address is obtained, the voter must have an address and a private and public key associated with the account to sign the transactions. To sign transactions, there are two types of accounts in Ethereum, according to the white and yellow paper [21]:

- EOA—Externally Owned Accounts—controlled by private keys and usually by wallets actioned by humans—Figures 5–7.
- Contract Accounts—controlled by their contract code.

**Figure 7.** Ethereum Address generation output.

Elliptic curve cryptography is used for Ethereum address generation and to validate the integrity of blocks chronologically in the blockchain.

Software wallets are used to store private and public keys or Ethereum addresses. In addition, a software wallet can be used to manage and carry out transactions. There are many more Bitcoin/Ethereum wallet types presented in [21–23] (Figures 6 and 7):

non-deterministic, deterministic, hierarchical deterministic, brain, paper-based, hardware, online, and mobile wallets.

Each transaction in the Ethereum network is saved into a database file on each full Ethereum node after the miners process it, creating Proof of Work consensus—Figure 8. The layers 1 and 2 of the **e-VoteD-App** solution have several components (see Figures 2, 4 and 8 for a better understanding of the entire architecture and data-flow):

- **ME—Layer 1**: Mobile Equipment of the voters for sharing the e-vote tokens, to have a token exchange protocol based on blind electronic signatures—see Figure 3 and Equations (1)–(3).

- **IoTeD—Layer 2**: IoT Embedded Device of the voters, which connects to the Ethereum Virtual Machine via Ethereum gateway and runs HD (Hierarchically Deterministic) wallet for e-voting process—see Annex A for the mathematical formalism.
- **PrivBN—Layer 2**: Private Blockchain Network:
  ○ Ethereum Virtual Machines linked to nodes and databases.
  ○ HD—Hierarchically Deterministic wallets.
  ○ Ethereum light-weight nodes—Mobile and IoT embedded devices.
  ○ Full Ethereum nodes—servers and e-voting tokens validation platform.
- **e-VUIDVP—Layer 1**: The electronic Voting UID Validation Platform, which checks the e-voting tokens validity and is a full node within the private Ethereum network.

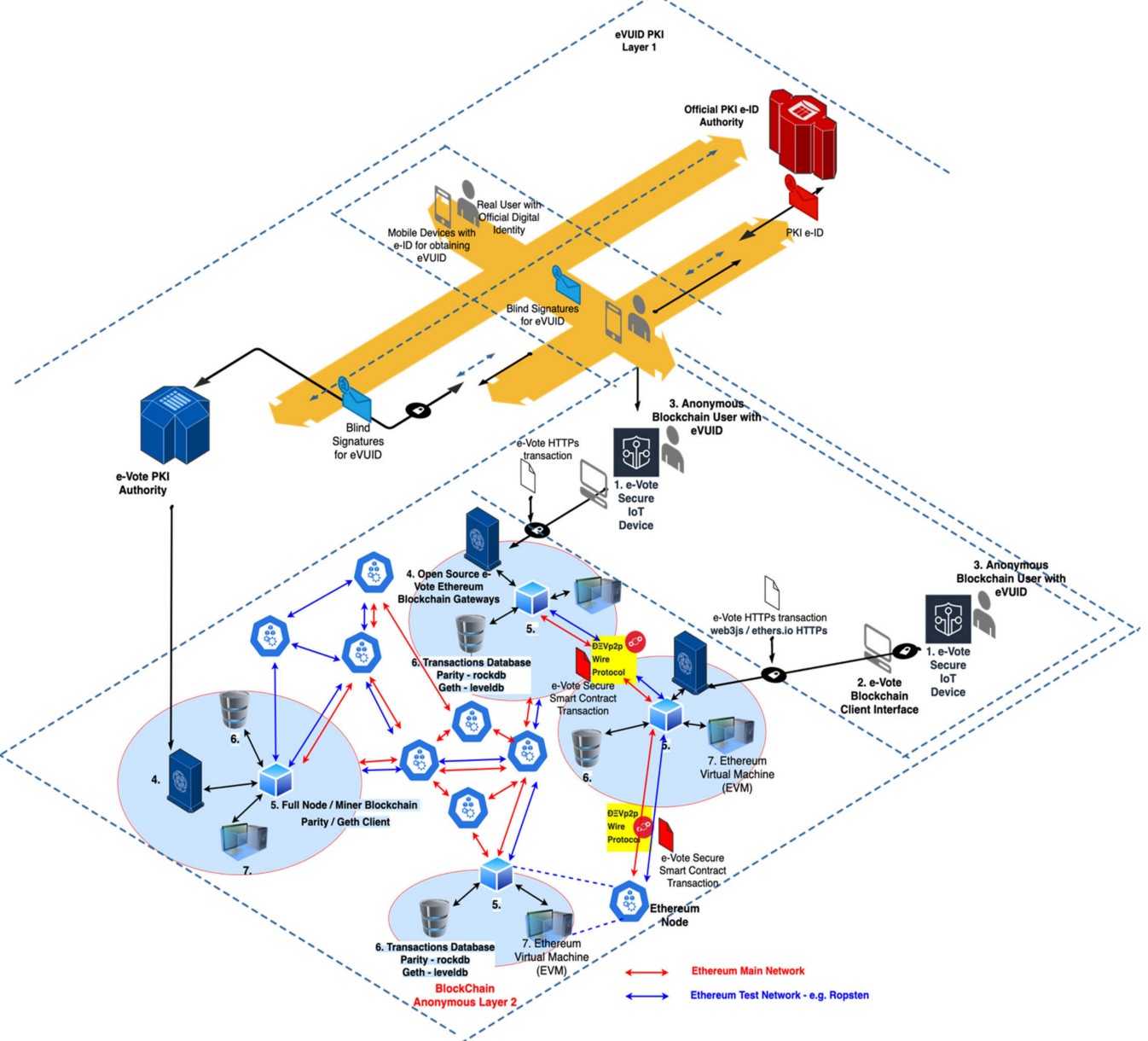

**Figure 8.** Detailed Layer 2 of the e-VoteD-App Solution based on Ethereum Blockchain and Smart Contracts running in EVM.

Taking a closer look into layer 2 of the architecture from Figure 8, we can identify the following entities:

1. e-Vote secure IoT device with a secure element, which runs the cryptographic functions for the ECDSA and securely stores the keys.
2. e-Vote blockchain client interface, which is a browser (e.g., Chrome, Firefox, Edge, Opera, etc.) extension/application that natively connects to the first component.
3. The voter who acts anonymously in the blockchain but presents a valid e-VUID blindly signed by the e-Vote Authority. In addition, the voter operates the first and second component to create the proper anonymous vote according to the smart contract. The vote is a special transaction that will be processed by the EVM and replicated in all blockchain nodes. Because the voter may not have the required hardware to run a full Ethereum node, an open-source library such as web3js or ethers.io can be used in order to submit the transactions to the 4-th component.
4. Open-source e-vote Ethereum blockchain gateway. For the PoC, the authors used infura.io gateway, but this is not open-source, although is free of usage with some certain limitations. This component must be open-source to avoid any concern regarding malicious behavior. The component is responsible for receiving the transactions via HTTPs interface but fully cooperating and running a full Ethereum node including the communication with the EVM.
5. The node is running a software Ethereum client such as Parity or Geth. The main Ethereum clients use two different database software solutions to store their tries. Parity, which is Ethereum's Rust client, uses **rocksdb**, whereas the Geth client and Ethereum's Go, C++ and Python clients all use **leveldb**. Each full-node client (Parity or Geth) manages the transactions, the state machine, and execution of the EVM—Ethereum Virtual Machine—for each transaction and contract by executing them into EVM.
6. Transaction database stores all transactions and is replicated by each **node** client (component 5) after Proof of Work consensus across the network using ÐΞVp2p Wire Protocol. As mentioned before, the node client Parity uses **rocksdb**, and Geth uses **leveldb**.
7. EVM—Ethereum Virtual Machine—executes the bytecode produced from the smart contracts written in Solidity or Vyper. The authors used a Solidity smart contract for the anonymous e-vote process.

A blockchain represents an ordered chain of chronological blocks. Each block from the blockchain is an aggregated set of data collected and processed by miners. The aggregated set of data must fit inside the block according to a set of validation rules. A cryptographic hash and a timestamp is associated with each block. Each new block added to the blockchain contains a cryptographic hash of the previous block. As such, all the blocks will make a chronologically ordered chain, uniquely generated starting with the so-called Genesis Block. An important set of the data within the block is represented by the transactions. A special transaction is used to create the smart contract for the e-vote process. For each vote, another special transaction is triggered by each voter in an anonymous way —see also Table 1 and Figures 8–10. Because an IoT/embedded device for voting may not have enough storage for the entire database of the transactions (approx. 700 GB in March 2021), Ethereum gateways are used—e.g., https://infura.io/ (accessed on 15 January 2022). The IoT device contains a secure element for the Ethereum wallet and runs an embedded web server to securely expose to the voter the web page for the vote. A special transaction is the one used for the smart contract creation, which has 0 Ethers for transfer but contains as data the bytecode for the voting smart contract.

Companies are relying more and more on blockchain technology to develop safer and more effective business. In order to understand better the Bitcoin and Ethereum technologies, important insights are given in [21–25]. The authors' e-VoteDApp solution is a standard D-App based on the Ethereum blockchain, and it uses blind signatures for distributing the voting tokens. The following sections reveal the proof-of-concept development of the secure and anonymous e-VoteDApp solution. In layer 2 of the e-Vote solution, the authors proposed an open-source Ethereum gateway as in Figure 7 and a

REST API compliant with node.js implementation for web3js and ethers.io—see Figure 9. This is a simple transaction from "account1" to "account2" for transferring 0.1 Ether. For this transaction, the "account1" must pay up to 21,000 units of gas at a unitary price of 10 gwei (1 ether = 1,000,000,000 gwei ($10^9$)).

**Table 1.** Ethereum Transaction Structure.

| Field Name | Field Length (Bytes) | Description |
|---|---|---|
| Nonce | Up to 32 bytes | A random number to prevent reply attacks |
| Gas Price | Up to 32 bytes | The price in Wei per "gallon" of gas to record the transaction in each database of the Ethereum nodes |
| Gas Limit | Up to 32 bytes | The maximum "gallons" of gas used for recording the transaction into the database |
| To | 20 bytes | EOA that is receiving ethers or contract address, which is in charge of the reception of the transaction that triggers running the e-vote smart contract in all EVMs of the full nodes. It could be zero for the contract creation. |
| Value | Up to 32 bytes | The value in Wei (subdivision of the Ether) for the transfer. It could be zero for the contract creation. |
| Data | 0 to unlimited | It can contain the ABI/bytecode for the contract execution. |
| V | 1 | Usually, 1 byte is the recovery ID in ECDSA. |
| r | 32 bytes | ECDSA r for the e-Vote signature. |
| s | 32 bytes | ECDSA s for the e-Vote signature. |

```javascript
const Web3 = require('web3')
const rpcURL = "https://ropsten.infura.io/v3/58238...ad39"
const web3 = new Web3(rpcURL)
var Tx = require("ethereumjs-tx").Transaction
const account1 = '0x3873dEaA1E8278f89FfB12F2aC28f3682079F1c3' // Your account address 1
const account2 = '0xe53429EBB0239065b4e2Ffb58F2e794D6E568C77' // Your account address 2
const privateKey1 = Buffer.from('b08ff6820178c5b900f86d...997bf1feab357a171bd0', 'hex')
const txData = {
    //nonce:   web3.utils.toHex(txCount),
    to:       account2,
    value:    web3.utils.toHex(web3.utils.toWei('0.1', 'ether')),
    gasLimit: web3.utils.toHex(21000),
    gasPrice: web3.utils.toHex(web3.utils.toWei('10', 'gwei'))
}
const sendRawTransaction = txData => (
  // get the number of transactions sent so far so we can create a fresh nonce
  web3.eth.getTransactionCount(account1).then(txCount => {
    const newNonce = web3.utils.toHex(txCount)
    const transaction = new Tx({ ...txData, nonce: newNonce }, { chain: 'ropsten' }) // or 'mainnet', 'rinkeby'
    transaction.sign(privateKey1)
    const serializedTx = transaction.serialize().toString('hex')
    return web3.eth.sendSignedTransaction('0x' + serializedTx)
  })
)
sendRawTransaction(txData).then(result => { console.log(result)}).catch(err => console.error(err))
web3.eth.getBalance(account1, (err,bal) => {
    console.log('account 1 balance :' , web3.utils.fromWei(bal, 'ether'))
})
```

```
Go 1.16.2 ⚡ ⊗ 0 ⚠ 14  <> Cloud Code   minikube

⌨  ⚙    cloudshell ✕   + ▾

account 1 balance : 4.577423403242096726
{
  blockHash: '0x33727db3a291918cc97104a8429db48f27d0b39de190b2554120d6064cbba14e',
  blockNumber: 9890038,
  contractAddress: null,
  cumulativeGasUsed: 2906109,
  from: '0x3873deaa1e8278f89ffb12f2ac28f3682079f1c3',
  gasUsed: 21000,
  logs: [],
```

**Figure 9.** Ethereum web3js/ethers.io API in node.js accessing the Ethereum Gateway for a simple transaction Type 1.

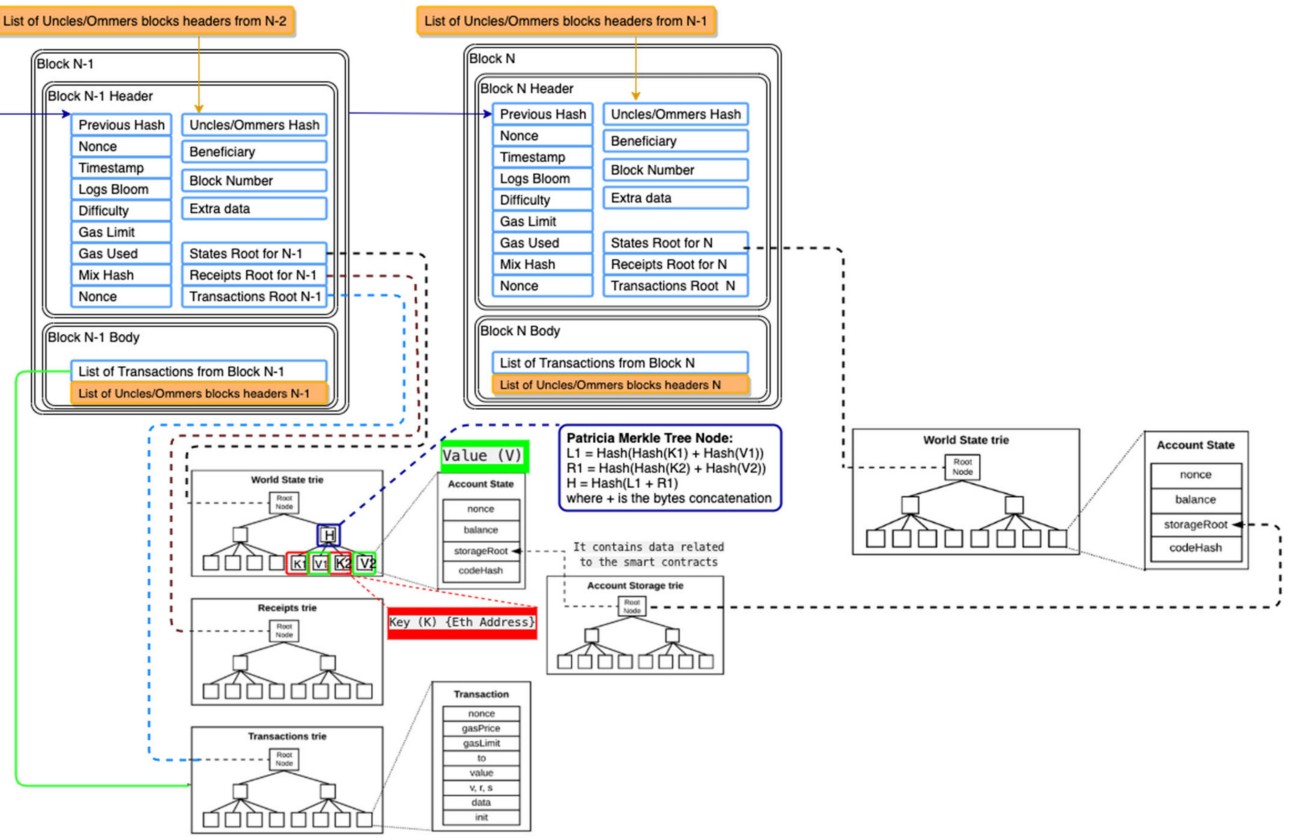

**Figure 10.** Ethereum block structure and the world state, receipts, and transactions in Patricia Merkle Tree(s)—tries.

As stated before, a transaction for a smart contract's creation must contain the "Value" of "zero", and the "Data" must contain the bytecode of the smart contract compiled with Solidity or Vyper. Each transaction is part of a block, and each block is chained by the others. The most simple analogy is with MS Excel/Google Sheets. These programs contain sheets (analogous to blocks), and each sheet has rows with certain columns (analogous to the transaction). A MS Excel/Google Sheets file is handled as a "database", and it should be replicated in each node from the blockchain node. Therefore, a problem will soon appear defined by whoever (which miner) wins the competition for inserting a transaction into the block from the blockchain, or in our analogy, for inserting an Excel row (transaction) into the Excel spreadsheet (block) from the Excel file (blockchain database) [26]. In the case of Ethereum blockchain, transactions include the following three types, which are also used in the solution (Table 1):

- Type 1: Transactions for transferring a value between two Ethereum addresses—EOAs (e.g, change the sender and receiver account balances)
- Type 2: Transactions for deploying a contract (e.g., creating an account, the contract account is an Ethereum address for the e-Vote smart contract)
- Type 3: Transactions for sending a message call to a contract (e.g., setting a value as vote in the smart contract by sending a message call that executes a setter method)

In Ethereum, according to the yellow paper, the block and the blockchain are more complicated than those of Bitcoin—please see both Figures 9 and 10 (technically, the source code for all types of transactions is the same as in Figure 9, but the payload details are little bit different).

As seen in Figure 9, each block from the blockchain has a header and a body. The block header includes general info such as previous block hash, nonce, and timestamp, but the block header also contains information that links the transaction, receipts, and the

world state as a Patricia Merkle tree data structure. The node from each tree is calculated as a hash function applied to the key-value tuple to maintain the integrity and fast search within the tree, as shown in Figure 9.

Let us assume that a transaction from the transactions tree contains as data the byte-code of the solution smart contract for e-voting. The smart contract for the e-vote was developed in Solidity and then compiled with SOLC to obtain the ABI (Application Binary Interface) and the bytecode, as shown in Figure 11.

```solidity
pragma solidity >=0.7.0 <0.9.0;

/**
 * @title Ballot
 * @dev Implements voting process along with vote delegation
 */
contract Ballot {

    struct Voter {
        uint weight; // weight is accumulated by delegation
        bool voted;  // if true, that person already voted
        address delegate; // person delegated to
        uint vote;   // index of the voted proposal
    }

    struct Proposal {
        // If you can limit the length to a certain number of bytes,
        // always use one of bytes1 to bytes32 because they are much cheaper
        bytes32 name;   // short name (up to 32 bytes)
        uint voteCount; // number of accumulated votes
    }

    address public chairperson;

    mapping(address => Voter) public voters;

    Proposal[] public proposals;
```

```
ABI:
[
    {
        "inputs": [
            {
                "internalType": "bytes32[]",
                "name": "proposalNames",
                "type": "bytes32[]"
            }
        ],
        "stateMutability": "nonpayable",
        "type": "constructor"
    },
...
]
Byte-code:
{
    "generatedSources": [],
    "linkReferences": {},
    "object": "6080604052348015610010576000080fd5b50604051610df4380380610df48339818101604052602081101561003357600080fd5b810190808051604
    "opcodes": "PUSH1 0x80 PUSH1 0x40 MSTORE CALLVALUE DUP1 ISZERO PUSH2 0x10 JUMPI PUSH1 0x0 DUP1 REVERT JUMPDEST POP PUSH1 0x40 MLOA
```

**Figure 11.** e-Vote Smart Contract generation form in Solidity language.

A transaction for creating a contract is just a standard transaction that has an empty address for the receiver. When a transaction for creating the contract is inserted into the blockchain by the miners, the transaction's data byte-array is seen as the EVM code. After executing the EVM, the value returned is used as the code for the new contract.

The full code that will end up being inserted into the blockchain to create the aforementioned name registry is presented in Figure 12 (let us assume that e-vote/ballot smart contract has the bytecode generated from Figures 12–14 instead of that from Figure 11 for the sake of simplicity):

One does not have to program in low-level assembly; a high-level language especially designed for writing contracts, known as Solidity, exists to make it much easier for one to write contracts (there are several others, too, including LLL, Vyper and Mutan), but in the next sections, there are highlights using the EVM bytecode for the sake of understanding.

An important characteristic of the EVM execution is that every single operation executed inside the EVM is done simultaneously on every full node. The Ethereum 1.0 consensus model relies on this approach, and it has the benefit that any EOA or contract address may call any other contract at almost zero cost, but the computational steps within the EVM are expensive. Therefore, there are two types of use cases for the smart contracts in terms of what is acceptable or unacceptable to include as the EVM bytecode/business logic:

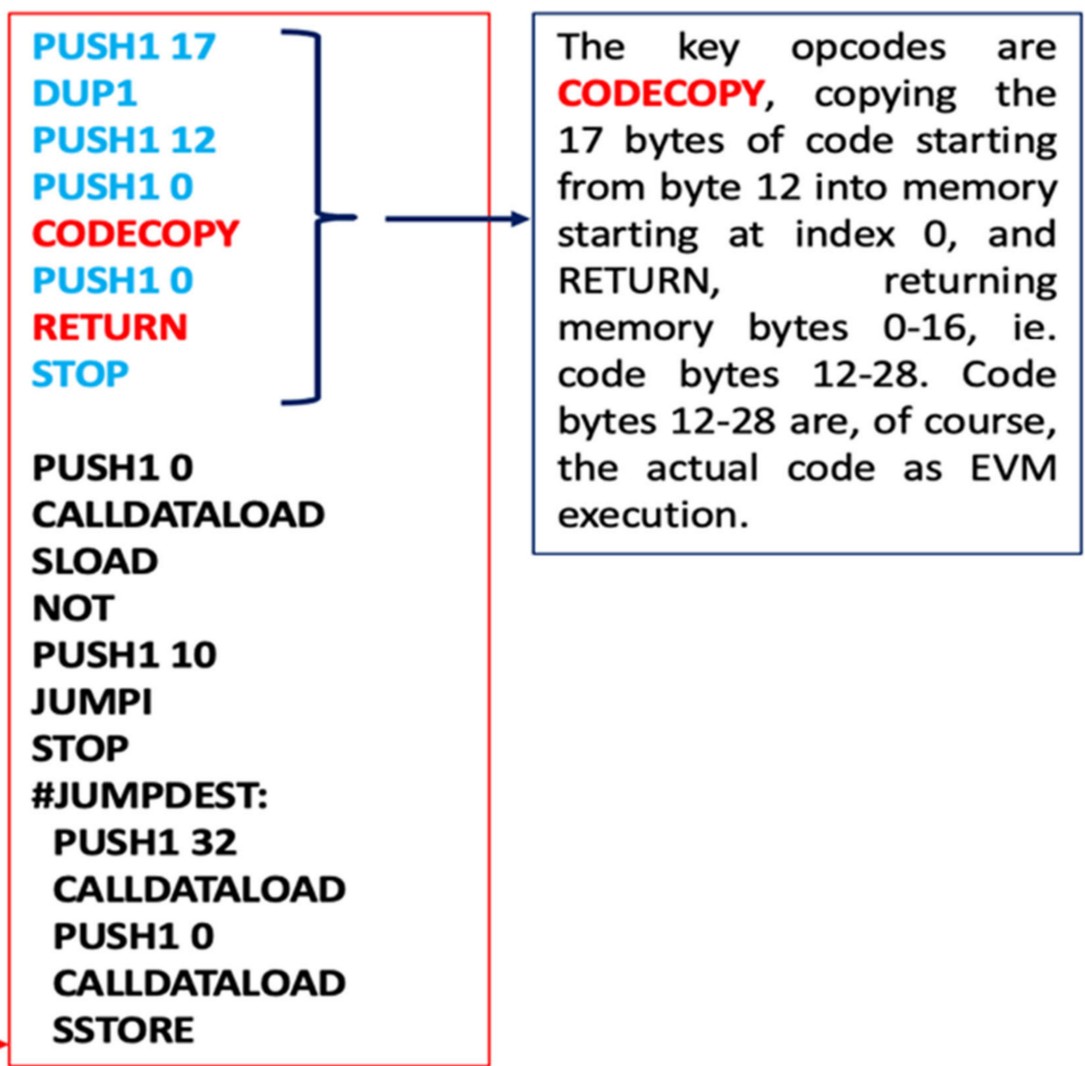

**Figure 12.** Ethereum Virtual Machine execution of the smart contract bytecode for deploying into the blockchain.

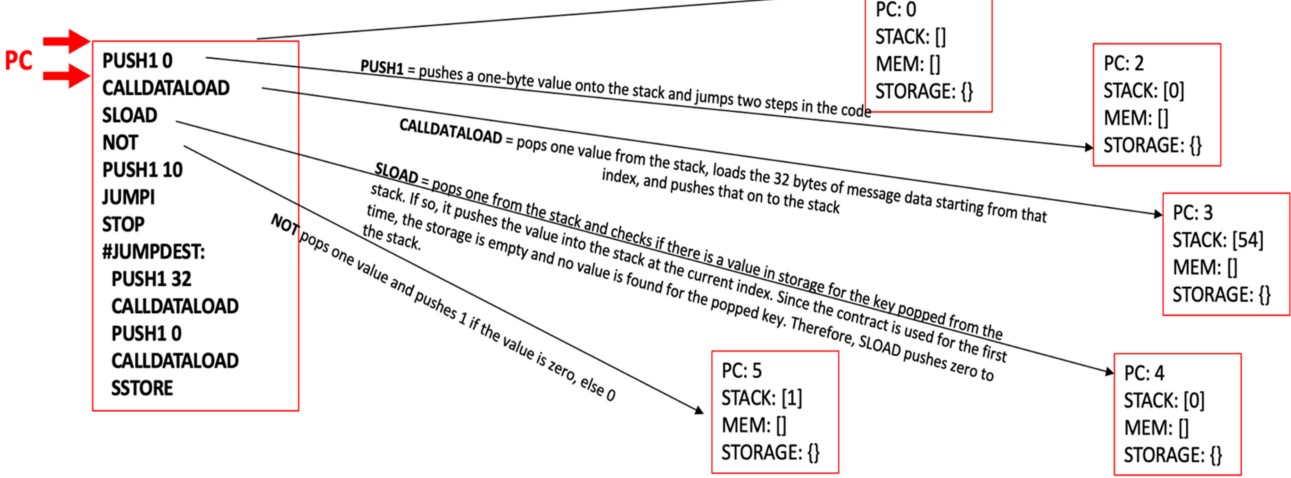

**Figure 13.** Ethereum Virtual Machine execution of the smart contract bytecode simple—sequence 1 (PC = Program Counter, MEM = Memory).

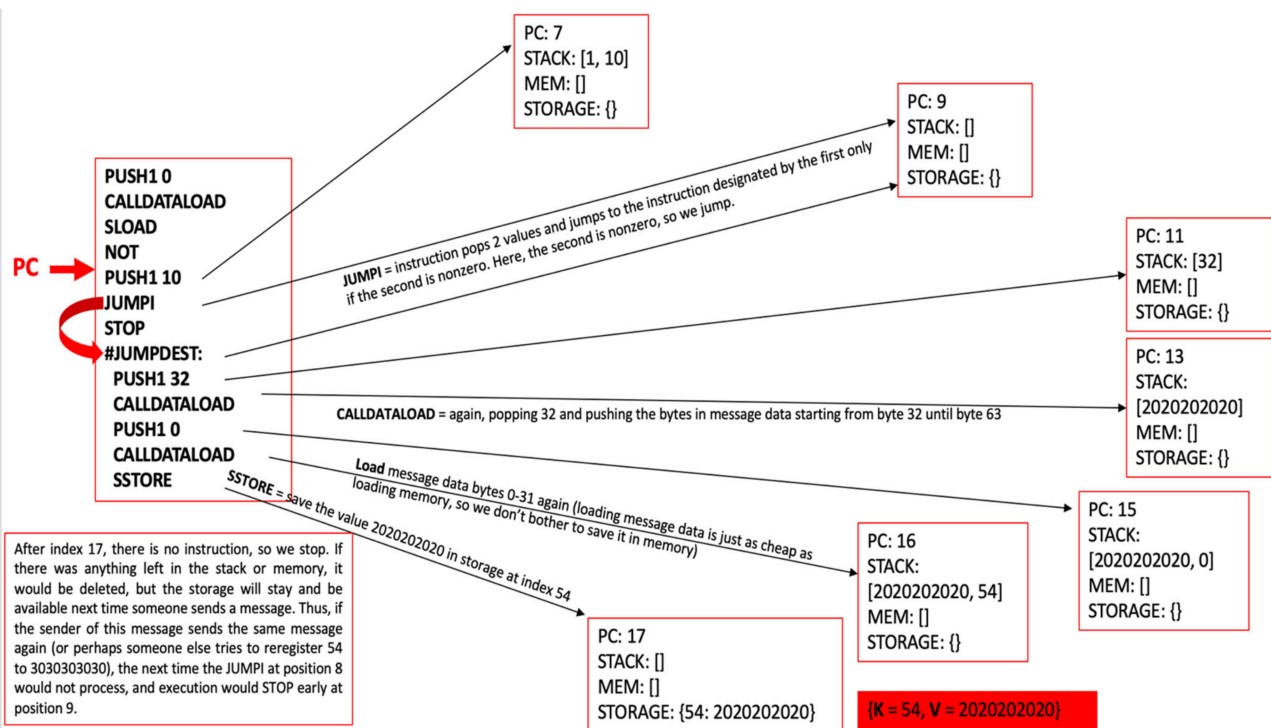

**Figure 14.** Ethereum Virtual Machine execution of the smart contract bytecode simple—sequence 2 (PC = Program Counter, MEM = Memory).

- Acceptable usage of the EVM includes running business logic (control structures as branching—if-then-else) and verifying signatures and other cryptographic operations;
- Unacceptable usage includes using the EVM as a Message-Oriented Middleware, e-mail or text messaging system, artificial intelligence genetic algorithms, graph analysis, or machine learning.

The Ethereum protocol has a fee per computational step, to prevent deliberate abuse. The fee is market-based, and it is established for the miners' competition in order to insert a transaction within a block from the blockchain; details about the economic reasoning and the block operation limit system are given in [21]. The fee mechanism and how it impacts the block header and the code is as follows (Figures 10 and 11):

- For each transaction there is a price quota associated, GASPRICE, and a transaction value, STARTGAS;
- The STARTGAS is the number of units assigned to the transaction, and the GASPRICE represents the fee paid for the transaction per each unit;
- When sending a transaction, the first operation during evaluation is to subtract START-GAS multiplied with the GASPRICE Wei in addition to the transaction's value from the sender's account balance. The GASPRICE is set by the sender of the transaction, but it can be refused by miners if GASPRICE value is too low.

After creation and deployment of bytecode for smart contracts through the entire blockchain, as the data part of transactions, database replication is started at least for the full nodes of the blockchain. The Ethereum execution is stateless, and the state of each account does not change. Nevertheless, any user can trigger an action and send a transaction into the blockchain from an externally owned account, EOA, starting the Ethereum's VM execution. If the destination is another EOA, then the crypto-currency Ether, ETH, is transferred. However, if the destination is a contract address, then the contract activates and automatically runs its code, in this case the e-Vote/Ballot smart contract. When a new transaction is targeting the smart contract address with a data field encapsulating the request call for a specific function from the e-voting smart contract,

then the bytecode is executed in every full node of the Ethereum blockchain within the EVM—Ethereum Virtual Machine. The smart contract code, running in EVM, can perform the following actions: modify internal storage, read the storage of received message, and send messages to other contracts, starting their execution in turn. Internal EVM processing is achieved by using a stack-based bytecode language, a mix between Java Virtual Machine bytecode and Lisp. An EVM program is a sequence of opcodes [27], as in Figures 13 and 14:

The contract is used as a name registry, since anyone can send 64 bytes of data, 32 for the key and 32 for the value. If the key is not registered, then the contract registers the value for that key location.

The initialization phase consists in setting the program counter, PC, with value 0 and clearing the memory and the 32 byte stack region. The memory is an infinitely expandable byte-array that is used for internal computations.

Once all of these elements are explained, it is easy to understand that in layer 2 of the e-VoteD-App solution (Figure 8), full Ethereum nodes are run to perform the e-voting process through smart-contract calls via the Ethereum transactions. The Ethereum transactions are secure and anonymous, triggering the execution of the smart contract byte code in all EVMs (Ethereum Virtual Machines) from the full nodes. In the next section, the conclusions are presented along with ways of improving the presented solution.

## 5. Implementation Issues and Results

For the proof of concept of developing **IoTeD** (Internet of Things Embedded Device) as a part of e-VoteD-App solution, the authors used a Raspberry Pi development board. This board is very good for proof-of-concept development, but for real devices in production, other specialized devices may be built, and different suitable ARM boards are used. For this embedded device, two approaches were used:

- The IoT device featured buttons in order to perform the vote —please see Figure 15;
- The IoT device could render and display a web page in a web browser from its internal web server—please see Figures 16 and 17.

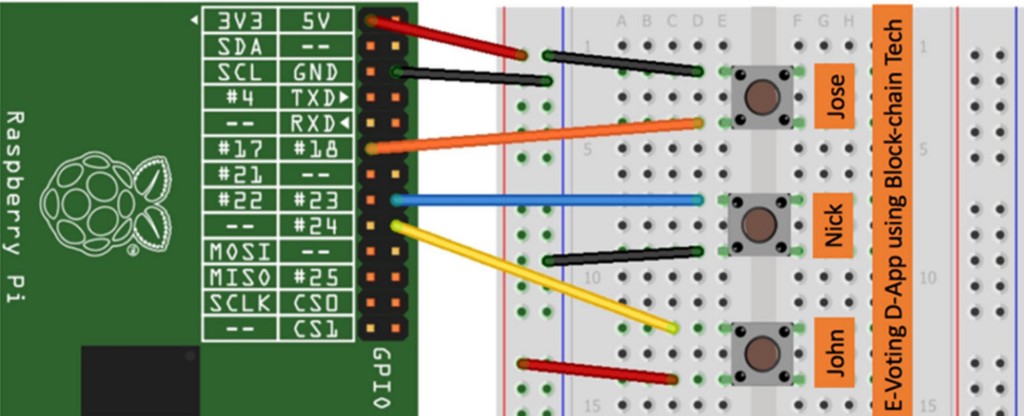

**Figure 15.** Hardware Diagram of IoTeD (Internet of Things embedded device) for the e-Voting D-App solution.

The Raspberry Pi implementation board was used for the e-voting D-App solution that is based on Ethereum blockchain technology. The connection to the board is made via GPIO pins; see Figure 15 for hardware schematics.

The software interactions for Figure 15 are presented in Figure 16, showing all the components involved for one entity, the voter who is using a board for the elections that has Ethereum node software running on top of it.

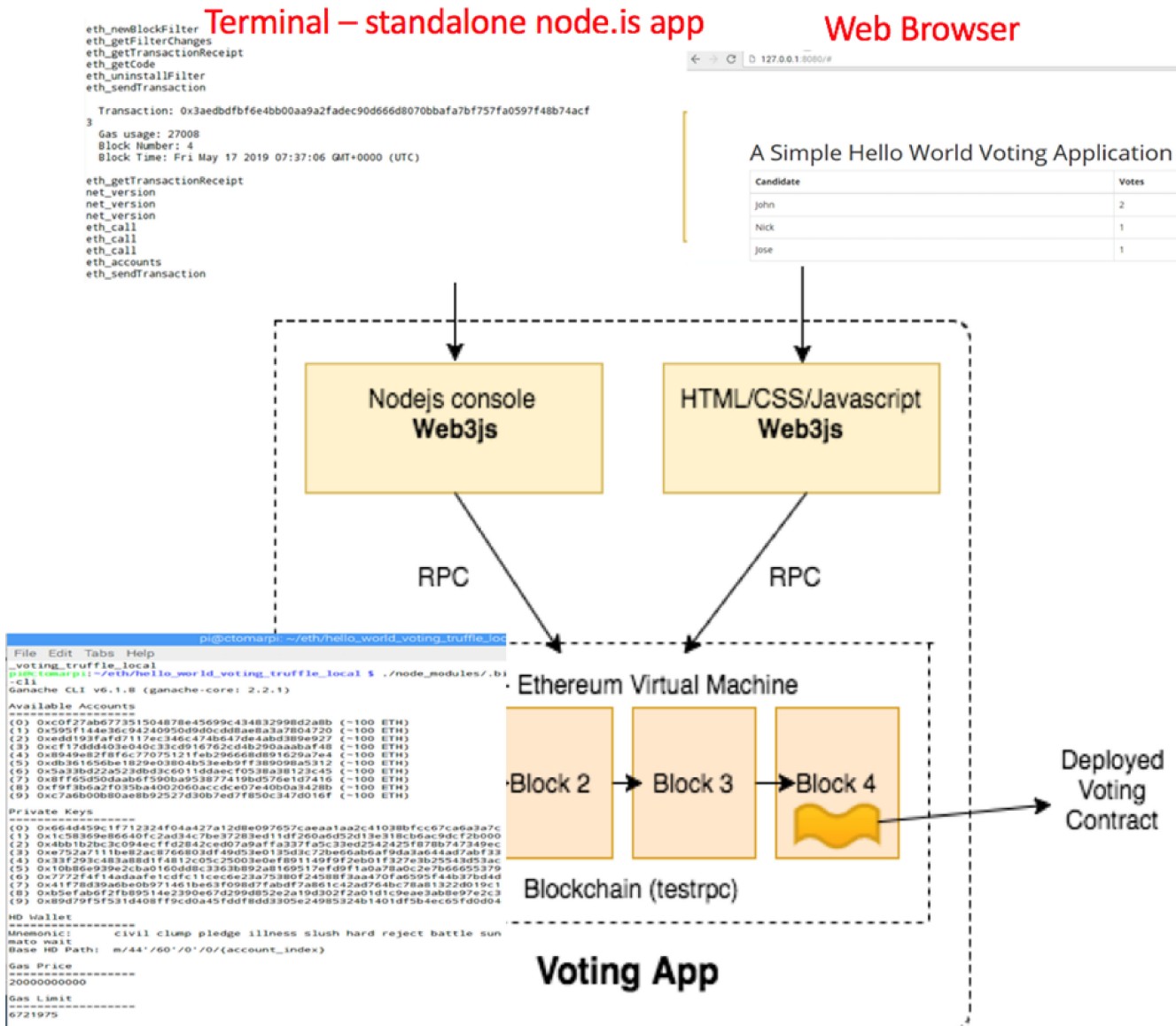

**Figure 16.** The web interface for displaying the votes number available for access on each IoTeD node of the e-voting D-App solution.

The top left of Figure 16 presents a terminal application that runs node.js in the embedded device. The voting events triggered in Figure 15 are captured by the application with the help of a Java device input/output library over general-purpose IO pin, followed by a remote call using JavaScript/node.js script that handles communications with the Ethereum VM (EVM). A software wallet is attached and connected to the EVM, capable of running smart contracts bytecode. After that, each vote triggered from Figure 15 represents a transaction that is stored in the local node of the embedded device. The entire blockchain is synchronized with all the nodes within the P2P network. Each voter uses an embedded device for the election process. Any voter may check at any moment the results of the voting process in an anonymous way and the number of votes for each candidate in the election process.

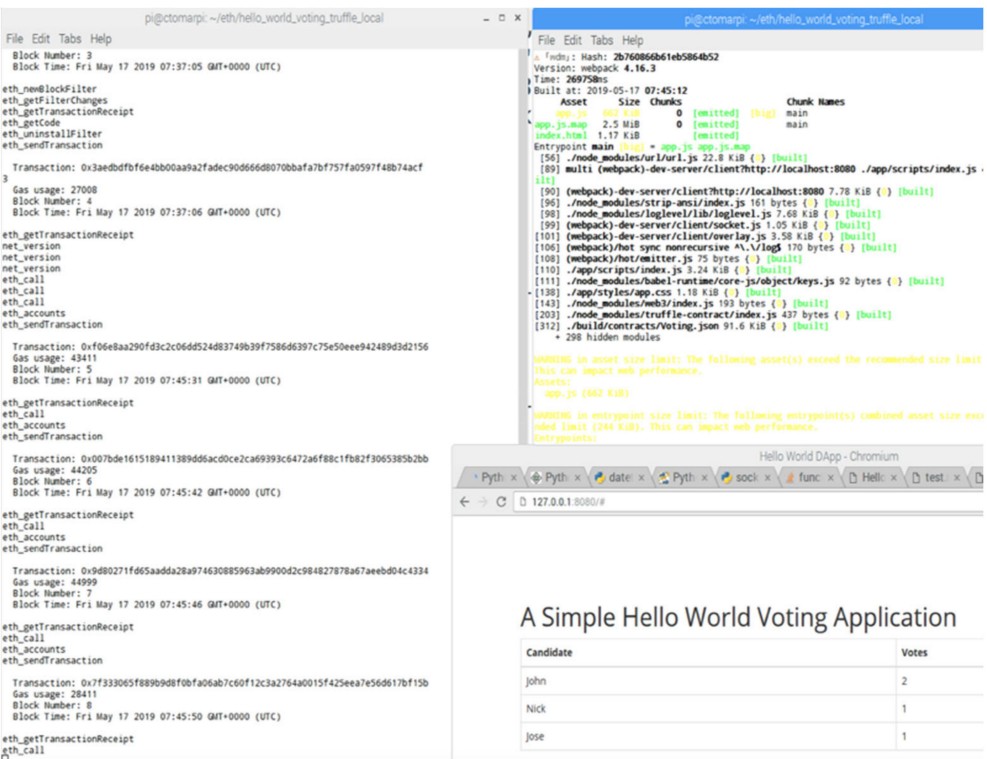

**Figure 17.** The web interface for displaying the votes number available for access on each IoTeD node of the e-Voting D-App solution, not necessarily involving physical buttons but only the web interface rendered by the embedded web server from the device.

The top right of Figure 16 presents the web browser results, which are detailed in Figure 17 by running locally in debug mode. Every vote started on the terminal triggers a transaction in the global blockchain. Repeated voting is avoided, because only one successful transaction per wallet address is accepted. The communication between entities can be verified against sensitive data leaks using [28]. Based on the AVISS protocol [29], the tool provides a modular and a formal language to describe sensitive data in the protocols used by the application, being able to output a thorough analysis using state-of-the-art automatic analysis techniques. In this stage of development, we did not use such systems, but we could actually benefit from those reports in order to show that this is a safe architecture.

In terms of security, each transaction is hashed, and then the hash is digitally signed. If there is no secure element, by default, the cryptographic algorithms run on the CPU of the embedded device, which is a security vulnerability. An improvement would be to run the sensitive code in a secure element (e.g., Java Card). An alternative to the embedded device is to use a mobile smart phone. From the mobile device, the option for a given candidate can be selected using a GUI application, and the vote transaction is hashed and digitally signed by the NFC chip/(U)SIM/eSIM secure element or an NFC external smart card, which are tamper-resistant and CC/FIPS EAL 4+ certified. If the secure element is an external NFC proximity smart card, then the communication between the mobile application and the smart card will be contactless via NFC/RFID.

The current solution is based on Ethereum blockchain, but other frameworks may be used. A comparison between frameworks is presented in Table 2:

Because the Bitcoin blockchain does not have a smart contracts feature, a e-voting system that uses Bitcoin would be challenging to implement. For this reason, the authors used Ethereum as the main blockchain platform support. The advantage of using Ethereum over Quorum or Exonum is that the platform is actively maintained and developed.

Comparing the characteristics of the existing e-voting solutions, the authors' solution is better in terms of anonymity (because of e-VUID and blind signature) and scalability—see Table 3.

**Table 2.** Blockchains Platform Comparison.

| Blockchain | Consensus Algorithm | Transactions per Second | Smart Contract Language | Platform Programming Language | Decentralized |
|---|---|---|---|---|---|
| Ethereum (Go/Parity) | PoW, PoS, and PoA | 15–30 (PoW) and up to 100 k (Ethereum 2.0) | Solidity/Vyper | Go/Rust, JavaScript | Yes |
| Quorum | QuorumChain, IBFT, and Raft-based consensus | Around 100 | Solidity | Go, C, JavaScript | Partially |
| Exonum | Custom-built BFT algorithm | Up to 5000 | Rust | Rust | Yes |
| Hyperledger Fabric | Decoupled in 3 phases: Endorsement, Ordering, Validation | Up to 3000 | Go/JavaScript/Java | Go | Yes |
| Bitcoin | PoW | 4–7 | N/A | C++ | Yes |

**Table 3.** e-Voting Solutions Comparison based on Characteristics.

| Characteristics/ Solutions | e-VoteD-App Proposed Solution (Ethereum) | Follow My Vote (Bitcoin) | Voatz (HyperLedger Fabric) | Polys (Ethereum) | Agora (Custom Blockchain Based on Bitcoin) | Lai, Hsieh [30] (Ethereum) |
|---|---|---|---|---|---|---|
| Anonymity | X (Based on e-VUID and blind signature) | X (Claimed but not described) | X (Claimed, with mix net based on homomorphic crypto, therefore not implemented) | X (Claimed based on blind signatures but not-described) | X (Claimed but not described) | X (Claimed but not implemented) |
| Audit | X | X | X | X | X | - |
| Accuracy | X | X | X | X | X | X |
| Integrity | X | X | X | X | X | - |
| Verifiability by Voter/Transparency | X | X | X | X | X | X |

The existing commercial solutions that use blockchain for e-voting do not describe how anonymity is achieved. The academic solutions for e-voting describe anonymity algorithms, but they are either hard to implement (e.g., usage of homomorphic functions) or lacking details about implementation.

The BOM (Bill of Materials) for the IoT devices used in e-voting POC includes:

- Raspberry PI or i.MX 7ULP EVK board~50 USD/200 USD
- Breadboard for Buttons/LCD~10 USD
- Buttons~5 USD or Display LCD-3.5″ 320 × 480~40 USD

## 6. Discussion and Conclusions

Such electronic voting systems [31] could be easily implemented, contributing to their early adoption in different areas of human activity. e-Voting could become the next major democratic instrument, a binder of e-democracy, as mobile and IoT technology steps into place.

The ubiquitous character of such technologies will allow individuals to easily adopt e-voting systems, so that anyone could vote and verify their election without compromising voters' privacy, ensuring state-of-the-art cryptography. As a future development, PQC—Post-Quantum Cryptography [32,33]—must also be considered in terms of public key algorithms, because RSA—Rivest–Shamir–Adleman—and ECDSA—Elliptic Curve Digital

Secure Algorithm—are not safe anymore. Therefore, as future work, the authors are aiming to use a blockchain that replaces the existing signature algorithms, ECDSA with a round 3 NIST algorithm, because of Post-Quantum Cryptography (PQC). This analysis will be developed based on the performance generated by the secure elements from Table 4:

**Table 4.** PQC Algorithms RAM and Processor minimal requirements suitable for Secure Elements.

| Scheme | Device | Chip | EEPROM (KB) | RAM (KB) |
| --- | --- | --- | --- | --- |
| McEllice (code-based) | SC Chip | 16 bit Infineon SLE76 | 310 | 4.5 |
| NTRU (lattice-based) | FPGA | 32 bit ARM7TDMI | 13.5 | 3.5 |
| Crystals-Kyber (lattice-based) | SC Chip | Infineon SLE76 | <500 | 16 |

In the literature, there are non-blockchain solutions, which are providing good results at the theoretical level, but they are not suitable yet for real implementations, because they use theoretical concepts such as cryptographic "random oracles" or IND-CCA—indistinguishability under chosen plaintext attack. Cryptographic random oracle [34] is an oracle (a theoretical black box) that responds to every unique query with a (truly) random response chosen uniformly from its output domain. For instance, according to [35], "some protocols replace the trusted authority by a set of authorities, and privacy is guaranteed if less than a threshold number of authorities are corrupt". To have validation, stronger security is required. The corrupt authorities that try to fake the result of vote counting must always be spotted. Non-Interactive Zero-Knowledge proofs (NIZKs) are used in e-voting protocols, to provide verifiability. The non-interactive design of the NIZKs allows third parties that do not participate in the protocol to verify the correctness of the vote counting. Therefore, universal verifiability is obtained with NIZKs. Additionally, instead of using blockchain technology, users can cast their votes using a non-interactive protocol—NIZKs. The disadvantage of NIZKs is that their security uses setup assumptions such as the common reference string (CRS) or the theoretical random oracle (RO) model, and this is less secure than a blockchain technology.

Even though there are many studies reflecting on how to implement electronic voting systems [36,37], with multiple resources available on the Internet that reveal how to develop an e-voting application that uses blockchain technology [38–41], there are still not many real implementations—especially in the embedded/IoT (Internet of Things) area. The ones that exist do not use embedded devices, being software-based implementations without the use of a secure element or even of a SoC architecture.

Additionally, the proposed PoC should be improved by using LCD instead of buttons or replacing the embedded IoT board with a smart phone, but the crypto used for layer 1 and 2 of the e-VoteD-App must be performed only into a tamper-resistant device such as SIM/e-SIM (e.g., Java Card secure element).

The proof-of-concept presented in this paper highlights the security and advantage of blockchain technology used in a e-voting application and proposes improvement by using Java Card for the blockchain HD wallet and a smartphone device for the blockchain node, which will run the application and the communications modules with other nodes from the proposed solution.

**Author Contributions:** Conceptualization, C.T. and C.B.; methodology, M.P.; software, C.T. and M.P.; validation, C.T., M.P. and M.D.; formal analysis, M.P.; investigation, C.C.; resources, C.B.; data curation, M.D.; writing—original draft preparation, C.T.; writing—review and editing, M.D.; visualization, C.C.; supervision, C.T.; project administration, C.B. All authors have read and agreed to the published version of the manuscript.

**Funding:** This research received no external funding.

**Acknowledgments:** This work was supported by Bucharest University of Economic Studies.

**Conflicts of Interest:** The authors declare no conflict of interest.

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
