# Peer review of "Secure and Anonymous Voting D-App with IoT Embedded Device Using Blockchain Technology"

_electronics, doi:10.3390/electronics11121895_

Round 1

Reviewer 1 Report

The paper presents the construction of a proof of concept for a distributed and decentralized e-voting application in an IoT embedded device employing blockchain technology. However, The paper presentation and structure is very confused and not reader-friendly even for security specialist. 

  • Please arrange the sections and improve the paper structure
  • What are your paper contributions over existing e-voting systems? 
  • How did your paper improve the e-voting system and where are the results of communication cost and communication cost using the blockchain? 
  • You supposed that your protocol for IoT devices and I didn't find any relation in the text or any example for calculations using IoT network or any simulations or results. 
  • The paper references are not arranged
  • Most of the figures are low-quality figures 
  • Why you are including the explaining of ECC cryptography and examples ( this makes the paper not a research article ( it looks like a report or script)
  • Where is your methodology, results, comparisons ??? 
  • The paper didn't provide any scientific contributions, you didn't show any performance or measurements for the proposed solutions 
  • Please include the ECC part as preliminaries, not a complete section ( anyone who works in security know these details, just reference it in the main file) 
  • Please try to reduce the unneeded parts and add some results, measuring of security level for the proposed e-voting system? comparing the proposed system with other related systems?
  • Verification of the security analysis using AVSPA tool 
  • The paper structure of any security paper must include ( definition of the problem, system model, proposed solution, security analysis, verification of the security, measuring of the performance, Discussion of results and conclusions). I didn't find the core of the paper. 
  • The paper missed the conclusions , discussions, and  results 

Author Response

Hello,

Please see our coments in the PDF file from the attachments.

Thank you!

Reviewer 2 Report

In this paper, the authors proposed the construction of a proof of concept for a distributed and decentralized e-voting application in an IoT embedded device employing blockchain technology. This solution ensures complete voter anonymity and end-to-end security for all entities participating in the 11 electronic voting process.

The solution and protocols provide two major properties: privacy and verifiability.

Figure -1 is a helpful summary presenting the contribution of the work.

The paper presents the secure e-voting schemes and implementation challenges of 80 proof of concept development for a voting distributed and decentralized application in an 81 IoT embedded device using the blockchain technology.

Figure-8 I think this could be a useful diagram, but it needs to be clarified

To improve this work, the authors need to cover the following issues:

  1. The writing style need to be refine, authors need review the paper for grammar errors: incomplete/ incorrect sentences
  2. The motivation and contribution are not clear and the details of the figures need to be improved.
  3. References need to be reviewed by the authors. Certainly, many other surveys exist which cover the fundamentals e-VoteD-app.

Author Response

(The authors gave the same response as above.)

Reviewer 3 Report

The paper "Secure and Anonymous Voting D-App with IoT Embedded Device using Blockchain Technology" presents an interesting topic. However, the paper needs more work before reaching the level of a journal paper.

The first issue is the use of the English language. Unfortunately, it is hard to understand several paragraphs, which distracts from the main idea.
Second, the authors expend too much space/time explaining concepts related to blockchain/smart contracts/cryptography that is not used later for highlighting the benefit or merit of their proposal. 
Third and most important, there is missing experimentation and thorough analysis that justifies why their proposal is better. More important, why this proposal is better than other blockchain-based systems. 
Given that the authors present a proof-of-concept, they should provide a performance analysis of the devices, the voting system costs and time, etc.  
The current version of the manuscript reads more like a report of a University-level course project than a research paper. 

Author Response

(The authors gave the same response as above.)

Round 2

Reviewer 1 Report

However, the authors made improvements to the paper; it is still not a friendly reader and lacks clarity. 

I am sorry I would like to reject this paper and encourage resubmission. 

The authors must arrange the sections, the idea, the results, and finally, the conclusions. 

Inserting your code parts in a research paper is not recommended, and you are required to discuss your results.  The references are not arranged. Also too many figures without good explanation or good caption. 

Snapshots from your code without clear discussion. 

However, I reread the paper content, but I didn't understand what the authors did. 

Author Response

Please see details in the PDF.

Reviewer 2 Report

The comments were addressed accordingly

Author Response

No revision need it, the comments have been addressed accordingly.

Reviewer 3 Report

The revised version of the paper "Secure and Anonymous Voting D-App with IoT Embedded Device using Blockchain Technology." presents significant improvements. This version of the manuscript is written better, and the authors present their idea more clearly. 
However, key elements still need to be addressed before publication. 

In the abstract (and title),  the authors mention "IoT embedded device employing blockchain technology."  However, their implementation uses an SoC board (i.e., Raspberry PI), whereas IoT embedded typically refers to Microcontrollers (i.e., STM, Arduino). This description can be misleading as truly embedded devices will add additional challenges to the voting system that would also be worth exploring. Authors should clarify this, particularly in the abstract. 

After reading section 1.1, it is unclear what is missing in current research. For instance, in reference [17], the authors state that "voters can cast a ballot anonymously". So, it is unclear what is missing from this solution already found in the literature ([17]) that motivates the version proposed by the authors that also guarantee anonymity. Overall, the authors need to identify the gap they are targeting clearly. 
Similarly, the paragraph on line 73 references several electronic voting systems (references [28-30]). However, the authors do not describe what is missing in those electronic voting systems. The related work should mention other works and provide a short analytical description (what is missing, could be improved, etc.), not just listing references. 
In line 87, "This solution provides full anonymity of the voters and ensures end-to-end security for the entities involved in the electronic vote process," the authors should add a sentence to summarize how they achieve that goal/claim.

Furthermore, on line 103, the paper states, "Figure 1 is a helpful summary presenting the contribution of the paper to the related area from which reader can get a better idea of how voters can interact with the voting system with total transparency from any security challenge." 
First,  the phrase "to the related area from which reader can get a better idea " is confusing. 
Second, and most importantly,  the figure shows a "Simplified overview of the voting processes" there is nothing of the author's proposal. So, it is impossible to extract the paper's contribution from that figure. 

Formulas and mathematical/formal content should all be presented in the same way, as in line 277 and not in line 283.
For completeness, authors should reference public-key cryptograph as it is done with ECC. 
Figure 5 does not show BIP 39 words for address generation; rather, it shows the voter process and the result of using Node JS to generate an Ethereum address from a private key. Some figures have incorrect labels and are not referenced properly in the text. Furthermore, Figures 5 and 7 have the same label. 
 The paragraph starting in line 344 is not clear. The authors should rephrase it better. The phrase starting in line 359 is quite confusing. What was used at the end? ECMAScript or OpenSSL? 

Considering the points previously mentioned, particularly the errors in labels and figures, the paper is still not on the level required for publishing.

Other minor comments:
In the abstract, "With privacy the paper solution ..." could be rephrased as "Regarding privacy, our solution ... " since "paper solution" could be interpreted as a traditional "paper" vote/ballot. 
Line 237 "There are many classes of the asymmetric algorithms" should be "There are many classes of asymmetric algorithms"

Author Response

Please see the PDF file.

Round 3

Reviewer 1 Report

I am advising the authors to rearrange the paper sections. Also, define the main problem, the solutions, methods, and results. The paper still looks like a report, not a scientific article. I am sorry for not recommending this paper for publication in its current form. 

Reviewer 3 Report

I don't see an updated version of the manuscript.

I've recieved the same version of the first round of revision, and also the authors response to that first revision.

Comments of my second review had not been addressed, at least, with the version I'm able to access

Author Response

Please see the attachment. The document includes also your previous suggestions. Last time it was an error when loading responses to your comments. Thank you for understanding!
